

# Reliability and accuracy of ultrasound image analyses completed manually *versus* an automated tool

Kealey J. Wohlgemuth[1], Malia N.M Blue[2] and Jacob A. Mota[1]

[1] Department of Kinesiology, University of Alabama - Tuscaloosa, Tuscaloosa, AL, United States
[2] Department of Exercise and Sport Science, University of North Carolina at Chapel Hill, Chapel Hill, NC, United States

## ABSTRACT

Analysis of Brightness-mode ultrasound-captured fascicle angle (FA) and fascicle length (FL) can be completed manually with computer-based programs or by automated programs. Insufficient data exists regarding reliability and accuracy of automated tools. Therefore, the purpose of this study was to determine the test-retest reliability of automatic and manual ultrasound analyses, while determining accuracy of the automatic tool against the manual equivalent. Twenty-three participants (mean ± SD; age = 24 ± 4 years; height = 172.2 ± 10.5 cm; body mass = 73.1 ± 16.1 kg) completed one laboratory visit consisting of two trials where vastus lateralis muscle architecture was assessed with ultrasound. Images were taken at both lower (10 MHz) and higher frequency (12 MHz). Images were analyzed manually in an open-source imaging program and automatically using a separate open-source macro function. Test-retest reliability statistics were calculated for automatic and manual analyses. Accuracy was determined with validity statistics and were calculated for automatic analyses. The results show that manual ultrasound analyses for FA and FL for both lower and higher frequency displayed good reliability ($ICC_{2,1}$ = 0.75–0.86). However, automatic ultrasound analyses for FA and FL revealed moderate reliability ($ICC_{2,1}$ = 0.61–0.72) for the lower frequency images and poor reliability ($ICC_{2,1}$ = 0.16–0.27) for higher frequency images. When assessed against manual techniques, automatic analyses presented greater total error (TE) and standard error of the estimate (SEE) for FA at lower frequency (constant error (CE) = −3.91°, TE = 5.57°, SEE = 3.45°) than higher (CE = −2.78°, TE = −4.54°, SEE = 2.45°). For FL, the higher frequency error (CE = 0.92 cm, TE = 2.12 cm, SEE = 1.15 cm) was similar to lower frequency error (CE = 1.98 cm, TE = 3.66 cm, SEE = 1.57 cm). The findings overall show that manual analyses had good reliability and low absolute error, while demonstrating the automated counterpart had poor to moderate reliability and large errors in analyses. These findings may be impactful as they highlight the good reliability and low error associated with manually analyzed ultrasound images and validate a novel automatic tool for analyzing ultrasound images. Future work should focus on improving reliability and decreasing error in automated image analysis tools. Automated tools are promising for the field as they eliminate biases between analysts and may be more time efficient than manual techniques.

Corresponding author
Jacob A. Mota, jamota@ua.edu

# INTRODUCTION

Brightness mode (B-mode) ultrasound is a popular imaging modality for assessing muscle size (*i.e.*, cross-sectional area, thickness) and architecture (*i.e.*, fascicle angle (FA), fascicle length (FL). Previous studies have shown that muscle structure which influence muscle size (*i.e.*, FL and FA) are important to whole muscle function and force production (*Lieber & Friden, 2000*; *Narici, 1999*). Specifically, muscle architecture can provide insight, *via* proxy, regarding the total amount of sarcomeres in parallel (*i.e.*, fascicle angle) and in series (*i.e.*, fascicle length) (*Maden-Wilkinson et al., 2020*; *Seynnes, de Boer & Narici, 2007*). Previous works have shown that in pennate muscles (*e.g.*, vastus lateralis, gastrocnemius), accounting for fascicles which are arranged at an angle relative to the tendon may be valuable as changes in fascicle angle can increase or decrease the mechanism of force generation within the muscle (*Kawakami, Ichinose & Fukunaga, 1998*; *Kwah et al., 2013*; *Narici, 1999*; *Seynnes, de Boer & Narici, 2007*). Therefore, assessing muscle size and architecture is important when assessing muscle function and force production and can easily be captured using B-mode ultrasound.

Analysis of B-mode ultrasound-captured FA and FL was classically performed manually using a protractor and printed image (*Kawakami, Ichinose & Fukunaga, 1998*), though contemporary computer-based image analysis options exist and are more frequently used (*Carr et al., 2021*). For instance, computer programs such as ImageJ (National Institutes of Health) or Photoshop (Adobe) include angle and straight line measurement tools which may be used to calculate FA and FL. Recently, automated analysis tools using various programs (*i.e.*, MATLAB, FIJI) used to estimate FA and FL have grown in interest (*Cronin et al., 2021*; *Seynnes & Cronin, 2020*). Automated tools can be more efficient as less time has to be devoted to training technicians and some automated scripts can bulk-process images instead of the individualized approached required by manual techniques. While the automatic image analysis programs may bring many benefits to future studies, insufficient data exists regarding reliability and accuracy of automatic ultrasound analysis tools compared to manual techniques. Also, there is not a clear consensus in regard to the ultrasound frequency that should be utilized as previous studies have used 10 MHz (*Carr et al., 2021*; *Ryan et al., 2016*), while 12 MHz is also common (*Mota & Stock, 2017*) and may uniquely impact image quality. Therefore, the primary purpose of this study was to determine the test-retest reliability of a novel, automatic ultrasound analysis tool for the assessment of vastus lateralis muscle architecture (*i.e.*, FA, FL). A secondary purpose of this study was to assess the accuracy of the novel, automatic ultrasound analysis tool against the manual analysis equivalent. Lastly, a tertiary purpose of this study was to examine the difference in reliability between images taken at a lower frequency (10 MHz) and higher frequency (12 MHz). It was *a priori* hypothesized that the novel, automatic ultrasound analysis tool for the assessment of muscle architecture would have good reliability ($ICC_{2,1} > 0.75$) and less error compared to the manual counterpart.

## METHODS

### Participants

Twenty-three participants (mean ± SD; age = 24 ± 4 years; height = 172.2 ± 10.5 cm; body mass = 73.1 ± 16.1 kg) were recruited from the local community for this study. Inclusion criteria consisted of individuals 18 to 35 years old and normal body mass index (BMI ≥ 30 $kg/m^2$). All participants read and signed an informed consent document explaining the risks and benefits of participating in the study. Participants were also asked to complete a health history questionnaire prior to participating in the study. This study was granted ethical approval by the University of Alabama Institutional Review Board (#21-02-4385) prior to data collection.

### Experimental design

Participants completed one laboratory visit (~1 h) which consisted of two independent data collection trials (trial 1, trial 2) separated by a 10-min rest period. During each trial, Brightness mode (B-mode) ultrasound was used to take images of the vastus lateralis (VL), where muscle architecture (*i.e.*, fascicle angle (FA) and fascicle length (FL)) of the VL was later assessed with open-source image analysis programs. During the 10-min rest period, participants remained relaxed and were not allowed to stand up or walk around the laboratory. The independent data collection trials consisted of identical methodologies and were subsequently used for the calculation of reliability statistics (see below).

### Ultrasonography

Participants laid supine on a portable exam table to undergo non-invasive imaging of the right VL using a B-Mode ultrasound imaging device (LOGIQ e R8; General Electric Company, Milwaukee, WI, USA) in conjunction with a multi-frequency linear-array probe (L4 – 12t - RS, 4.2–13 MHz, 47.1 mm field of view; General Electric Company, Milwaukee, WI, USA). The VL was marked at the proximal and distal musculo-tendon junctions as determined *via* ultrasound and length was measured with a flexible tape measure. To capture muscle architecture, a longitudinal image was taken by scanning the entire length of the VL (*i.e.*, proximal to distal musculo-tendon junction) while using the extended field of view function. Ultrasound gain and depth settings were held constant (*i.e.*, gain = 52 dB and depth = 6 cm) for each participant. Ultrasound frequency was systematically changed from 10 to 12 MHz (*i.e.*, lower frequency and higher frequency, respectively). Images were systematically taken in the following order (i) frequency 10 MHz and depth 6 cm and (ii) frequency 12 MHz and depth 6 cm. Caution was taken to ensure consistent and minimal pressure was applied to the probe as it moved across the skin to not compress the underlying muscle tissue. A generous amount of acoustic coupling gel was applied to the skin to enhance image acquisition. During data collection, a minimum of two quality images were taken per ultrasound frequency. A quality ultrasound image is defined as an image where the superficial and deep aponeuroses of the muscle of interest are clearly seen as well as fascicle that is clearly captured from superficial to deep aponeuroses. During offline-image analysis (see-below) the single clearest image was used for processing.

## Manual ultrasonography analysis

All images were manually analyzed in an open-source imaging computer program (ImageJ; National Institutes of Health, Bethesda, MD, USA), after scaling from pixels to centimeters. The angle tool in ImageJ was used to measure FA by tracing a clearly visible fascicle the distance between the superficial and the deep aponeuroses. The angle of interest was defined as the angle formed by the fascicle meeting the deep aponeurosis. To measure FL, ImageJ's straight-line tool was carefully placed on top of the previously described FA line to measure the length of the fascicle of interest (Fig. 1). During each measurement, the investigator carefully zoomed in to ensure the appropriate fascicle components were assessed. Both FA and FL measurements were recorded.

## Automatic ultrasonography analysis

The automatic ultrasound analyses were completed using the Simple Muscle Architecture (SMA) 17 macro function in FIJI (ImageJ; National Institutes of Health, Bethesda, MD, USA). The settings for SMA were set to the guidelines recommended by the original authors (*Seynnes & Cronin, 2020*). The author's programming code for the SMA function is written in ImageJ macro language and is run in FIJI. FIJI is an enhanced version of the popular open-source imaging platform, "ImageJ" (*Schneider, Rasband & Eliceiri, 2012*). The SMA function works by four distinct processes: (i) detect field of view, (ii) detect the superficial and deep aponeuroses, (iii) measure fascicle orientation, (iv) calculate parameters. This automated process then presents the operator with FA and FL, which were recorded. For full details on this algorithm, the reader is guided to the original work of *Seynnes & Cronin (2020)*. The automatic analyses were completed on the same images that were analyzed manually; however, if the SMA function could not locate the FA and FL using the image selected by the authors, the next best image was used until the SMA function correctly identified a FA and FL. This only occurred for one participant during trial 2, for which the SMA function could not locate a FA and FL for any image with a lower frequency. The SMA function was able to successfully analyze images for the other 22 participants for both trial 1 and 2.

## Statistical analyses

Test-retest reliability statistics (*i.e.*, intraclass correlation coefficient model 2, 1 ($ICC_{2,1}$), standard error of measure expressed as a percentage of the mean (SEM%), and the minimal difference (MD) needed to be considered real) were calculated between trial 1 and 2 for both the automatic and manual analyses of FA and FL, respectively. Calculations were performed according to test-retest reliability procedures described by *Weir (2005)*. Reliability ratings were determined by the ranges described by *Koo & Li (2016)*, that describe ICC estimates less than 0.5 as poor, 0.5 to 0.75 as moderate, 0.75 to 0.9 as good and 0.9 and greater as excellent reliability. To assess accuracy of the automated tool, validity statistics (*i.e.*, constant error (CE), total error (TE) and standard error of the estimate (SEE)) and Bland-Altman plots were calculated for the automatic analyses using the manual analyses as criterion values for both the FA and FL images taken with a lower and higher frequency during trial 1. The error calculations were completed as follows:

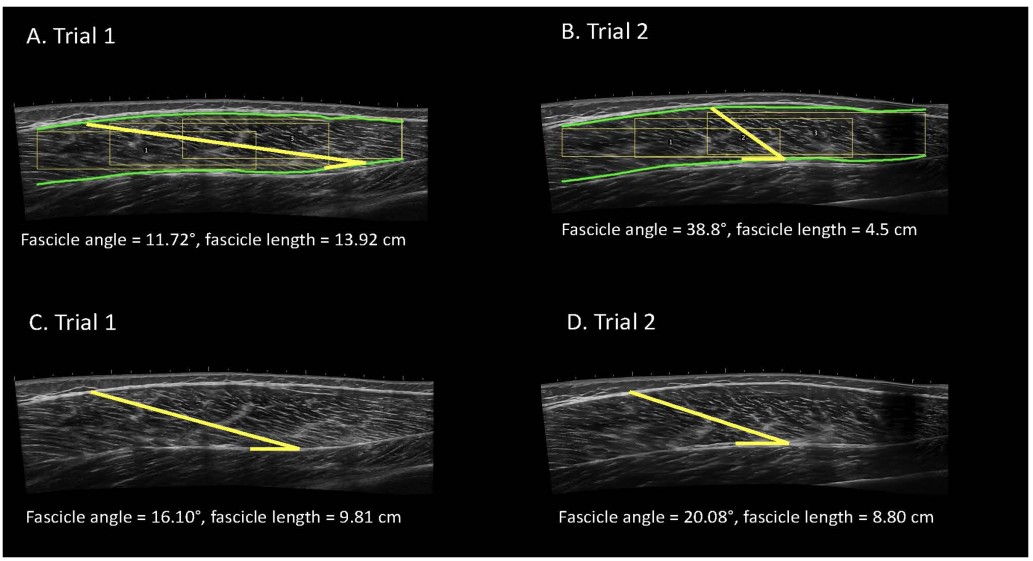

A. Trial 1

Fascicle angle = 11.72°, fascicle length = 13.92 cm

B. Trial 2

Fascicle angle = 38.8°, fascicle length = 4.5 cm

C. Trial 1

Fascicle angle = 16.10°, fascicle length = 9.81 cm

D. Trial 2

Fascicle angle = 20.08°, fascicle length = 8.80 cm

**Figure 1 Examples of automatic (A, B) and manual (C, D) analyses of the vastus lateralis completed by the same analyst for the same participant with the image taken at a higher frequency.** Participants completed one laboratory visit which consisted of two independent data collection trials. Images were systematically taken with various frequencies. Images were manually analyzed using ImageJ and the automatic ultrasound analyses were completed using a macro function in FIJI.

constant error (CE = criterion (manual analysis) – comparison (automated)), total error (TE = $\sqrt{\sigma}$ (comparison-criterion)$^2$/n), and standard error of the estimate (SEE = $\sqrt{\sigma}$ (comparison-criterion) $\times$ $\sqrt{1 - r^2}$). Statistical analyses were completed using R version 4.1.0 and RStudio (Version 1.4.1717) using the "blandr", "irr", and "psych" packages. The *a priori* criterion for significance was set at $\alpha = 0.05$.

# RESULTS

## Reliability of manual ultrasonography analyses

Manual ultrasound analyses for FA with a lower frequency had good reliability ($ICC_{2,1} = 0.75$, Cohen's $d = 0.05$) and the SEM% and MD were 13.69% and 7.40°, respectively. For FL with the same image settings, the manual analyses also had good reliability ($ICC_{2,1} = 0.83$, Cohen's $d = 0.14$) with the SEM% of 12.74% and the MD of 2.48 cm. The manual ultrasound analyses with a higher frequency had good reliability for FA ($ICC_{2,1} = 0.75$, Cohen's $d = 0.03$) with a SEM% of 9.99% and MD of 5.38°. For FL with the same image settings, the manual analyses had good reliability ($ICC_{2,1} = 0.86$, Cohen's $d = 0.02$) with the SEM% of 10.72% and MD of 2.16 cm. Further, the manual analyses for both FA and FL had good reliability with low absolute error. The reliability statistics for the manual ultrasound analyses are presented in Table 1.

## Reliability of automatic ultrasonography analyses

Automatic ultrasound analyses for FA with a lower frequency had moderate reliability ($ICC_{2,1} = 0.61$, Cohen's $d = 0.01$) and the SEM% and MD were 18.57% and 8.79°, respectively. For FL with the same image settings, the manual analyses also had moderate

**Table 1 Test-retest reliability of manual ultrasound analyses for the vastus lateralis.** Participants completed one laboratory visit which consisted of two independent data collection trials. Images were systematically taken with various frequencies. Images were manually analyzed using ImageJ and the automatic ultrasound analyses were completed using a macro function in FIJI. ($ICC_{2,1}$, Intraclass correlation coefficient $model_{2,1}$; SEM%, standard error of measure expressed as a percentage of the mean; MD, minimal difference values needed to be considered real).

| | | $ICC_{2,1}$ | Cohen's $d$ | SEM(%) | MD |
|---|---|---|---|---|---|
| Manual Analysis | Fascicle angle | | | | |
| | 10 mHz | 0.75 | 0.05 | 13.69 | 7.40° |
| | 12 mHz | 0.75 | 0.03 | 9.99 | 5.38° |
| | Fascicle length | | | | |
| | 10 mHz | 0.83 | 0.14 | 12.74 | 2.48 cm |
| | 12 mHz | 0.86 | 0.02 | 10.72 | 2.16 cm |
| Automatic Analysis | Fascicle angle | | | | |
| | 10 mHz | 0.61 | 0.01 | 18.57 | 8.79° |
| | 12 mHz | 0.27 | 0.18 | 25.59 | 12.12° |
| | Fascicle length | | | | |
| | 10 mHz | 0.72 | 0.05 | 27.96 | 6.13 cm |
| | 12 mHz | 0.16 | 0.23 | 31.38 | 6.88 cm |

reliability ($ICC_{2,1}$ = 0.72, Cohen's $d$ = 0.05) with the SEM% of 27.96% and the MD of 6.13 cm. The automatic ultrasound analyses with a higher frequency had poor reliability for FA ($ICC_{2,1}$ = 0.27, Cohen's $d$ = 0.18) with a SEM% of 25.59% and MD of 12.12°. For FL with the same image settings, the automatic analyses also had poor reliability ($ICC_{2,1}$ = 0.16, Cohen's $d$ = 0.23) with the SEM% of 31.38% and MD of 6.87 cm. Taken together, the automatic analyses for both FA and FL had poor to moderate reliability with moderate absolute error. The reliability of the automatic ultrasound analyses are presented in Table 1.

## Accuracy of automatic ultrasonography analyses

The automatic analyses for FA images taken with a lower frequency, had a CE of −3.91°, TE of 5.57°, and SEE of 3.45°. For FL with the same image settings, the CE was 1.98 cm, TE was 3.66 cm, and SEE was 1.57 cm. The automatic ultrasound analyses of FA for images taken with a higher frequency, had a CE of −2.78°, a TE of −4.54°, and SEE of 2.45°. For FL with the same image settings, the CE was 0.92 cm, TE was 2.12 cm, and the SEE was 1.15 cm. The validity statistics for the automatic ultrasound analyses are presented in Table 2. Accuracy at the individual level is presented in Bland-Altman plots (Fig. 2) for FA and FL with images taken at both lower and higher frequencies. The regression lines presented in the Bland-Altman plots were statistically significant, indicating proportional biases for higher frequency FA ($R^2$ = 0.22; $P$ = 0.03), higher frequency FL ($R^2$ = 0.34; $P$ = <0.01), and lower frequency FL ($R^2$ = 0.36; $P$ < 0.01), but not lower frequency FA ($R^2$ < 0.01; $P$ = 0.93).

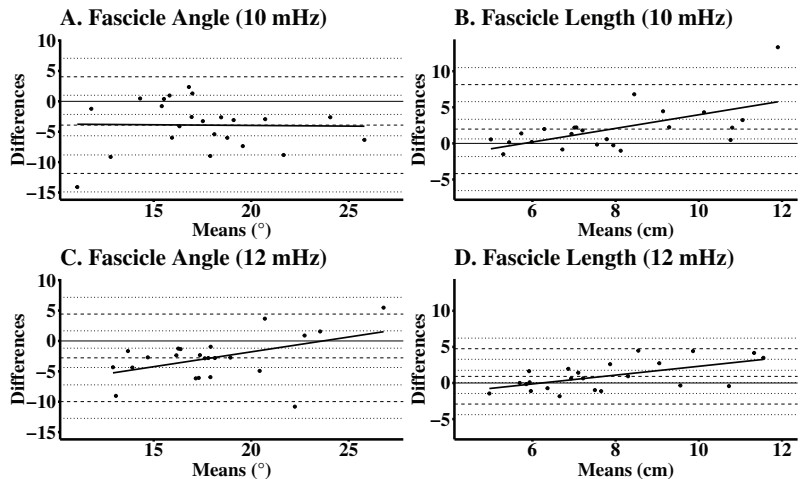

**Figure 2 Bland-Altman plots for automatic analyses of the vastus lateralis, while using manual analyses as the criterion.** Participants completed one laboratory visit which consisted of two independent data collection trials. Images were manually analyzed using ImageJ and the automatic ultrasound analyses were completed using a macro function in FIJI. Differences between automated and manual analyses for (A) lower frequency FA (mean difference (mdiff) = −3.91°, limits of agreement (LOA) = −11.90° to 4.04°); (B) lower frequency FL (mdiff = 1.98 cm, LOA = −4.18 to 8.14 cm); (C) higher frequency FA (mdiff = −2.78°, LOA = −9.98° to 4.42°); and (D) higher frequency FL (mdiff = 0.92 cm, LOA = −2.92 to 4.75 cm).                              

## DISCUSSION

This study examined the test-retest reliability of a novel, previously published, automatic ultrasound analysis tool for the assessment of VL muscle architecture (*i.e.*, FA, FL). In addition to assessing reliability, this study examined the accuracy of a novel, automatic ultrasound analysis tool by using manual ultrasound analyses as the criterion. The findings from this study show that manual ultrasound analyses for FA and FL for both lower and higher frequency displayed good reliability ($ICC_{2,1}$ = 0.75–0.86) and low absolute error (SEM% = 9.99–13.69%). The MD for manual analyses for FL ranged from 2.16 to 2.48 cm, while the MD for the FA ranged from 5.38° to 7.40°. However, automatic ultrasound analyses for FA and FL revealed moderate reliability ($ICC_{2,1}$ = 0.61–0.72) for the lower frequency images, poor reliability ($ICC_{2,1}$ = 0.16–0.27) for higher frequency images, and moderate absolute error for both lower and higher frequency images (SEM % = 18.75–31.38%). The MD for FA ranged from 8.79° to 12.12°, while the MD for FL ranged from 6.13 to 6.87 cm. When validated against manual techniques, automatic analyses presented greater TE and SEE for FA at lower frequency (CE = −3.91°, TE = 5.57°, SEE = 3.45°) than at higher frequency (CE = −2.78°, TE = −4.54°, SEE = 2.45°). For FL, the higher frequency error (CE = 0.92 cm, TE = 2.12 cm, SEE = 1.15 cm) was similar to the lower frequency error (CE = 1.98 cm, TE = 3.66 cm, SEE = 1.57 cm). These findings add to an emerging body of literature examining automatic ultrasound analysis tools, specifically the accuracy and reliability of an automated tool when assessing the muscle architecture of the vastus lateralis.

**Table 2 Validity of automatic *vs* manual analyses of the vastus lateralis.** Participants completed one laboratory visit which consisted of two independent data collection trials. Images were systematically taken with various frequencies. Images were manually analyzed using ImageJ and the automatic ultrasound analyses were completed using a macro function in FIJI. Validity of the automatic analyses were calculated using the manual analyses as criterion values for both FA and FL images taken during trial 1.

|  | Constant error | Total error | Standard error of the estimate |
|---|---|---|---|
| **10 mHz** |  |  |  |
| Fascicle angle | −3.91° | 2.12° | 1.15° |
| Fascicle length | 1.98 cm | 3.66 cm | 1.57 cm |
| **12 mHz** |  |  |  |
| Fascicle angle | −2.78° | 4.54° | 2.45° |
| Fascicle length | 0.92 cm | 2.12 cm | 1.15 cm |

## Reliability of ultrasound analytical techniques

Brightness mode ultrasound is a reliable tool for measuring skeletal muscle morphology (*Carr et al., 2021*; *Van Hooren, Teratsias & Hodson-Tole, 2020*; *Kwah et al., 2013*; *May, Locke & Kingsley, 2021*; *Rosenberg et al., 2014*; *De Souza Silva et al., 2018*). In the current study, manual ultrasound analyses indicated good reliability when assessing muscle architecture (*i.e.*, FA and FL) ($ICC_{2,1}$ = 0.75–0.86) with low absolute error (SEM % = 9.99–13.69). Two separate systematic reviews (*Van Hooren, Teratsias & Hodson-Tole, 2020*; *Kwah et al., 2013*), which contain similar studies, suggested ultrasound assessed FA and FL of the VL demonstrated moderate to excellent reliability (ICC = 0.51–1.0) with low absolute error (SEM% = 4.3–14.2). Recently, another study examined ultrasound assessed muscle architecture, but of the gastrocnemius medialis (GM) and gastrocnemius lateralis (GL) and reported moderate to excellent reliability (ICC: GM = 0.63–0.91, GL = 0.63–0.82) (*May, Locke & Kingsley, 2021*). Furthermore, automatic ultrasound analysis tools, have been recently developed (*Cronin et al., 2021*; *Seynnes & Cronin, 2020*); however, the literature evaluating the reliability of the automatic programs are sparse. In the current study, automatic ultrasound analysis reliability for muscle architecture (*i.e.*, FA, FL) were poor to moderate ($ICC_{2,1}$ = 0.16–0.72) with moderate absolute error (SEM% = 18.57–31.38). *Cronin et al. (2021)* reported reliability statistics of a semi-automated tracing of ultrasound images of the biceps femoris muscle architecture and reported a low coefficient of variation (CV) for the semi-automatic tracing of FA (CV% = 2.58–10.70) and FL (CV% = 0.64–1.12). It is likely that the differences between the previous studies and the present work are due to a difference in muscle architecture (*i.e.*, FA, FL) assessment procedures. The present work utilized a single muscle fascicle for each technique (*i.e.*, auto, manual), whereas many other studies (*Akagi, Hinks & Power, 2020*; *Gerstner et al., 2017*; *Narici et al., 2021*; *Sarto et al., 2021*) use the average of multiple fascicles. It is likely that taking an average of multiple fascicles is superior to assessing just a single fascicle. However, the use of the SMA tool in the present study only incorporates a single muscle fascicle into its muscle architecture measure (*i.e.*, FA, FL) which was mimicked in the manual technique used in the present work.

## Accuracy of ultrasound analytical techniques

In the present study, automatic ultrasound analyses were validated against the aforementioned manual technique, resulting in low error for FA (CE = −3.91° to −2.78°, TE = −4.54° to 5.57°, SEE = 2.45–3.45°) and FL (CE = 0.92–1.98 cm, TE = 2.12–3.66 cm, SEE = 1.15–1.57 cm) calculations across ultrasound frequencies. Previous studies have discussed the use of ultrasound imaging as a valid modality for measuring phantom or cadaver muscle architecture (*Bénard et al., 2009*; *Van Hooren, Teratsias & Hodson-Tole, 2020*; *Kawakami, Abe & Fukunaga, 1993*; *Kellis et al., 2009*; *Kwah et al., 2013*). In some of the previous works (*Van Hooren, Teratsias & Hodson-Tole, 2020*; *Kwah et al., 2013*) which report validity, traditional validity metrics (*i.e.*, CE, TE, SEE) were not employed. The current study reported CE, TE, and SEE to evaluate the error in the automated measure of muscle architecture (*i.e.*, FA, FL) compared to manual techniques, while other studies used coefficient of multiple correlation or presented reliability statistics (*i.e.*, ICC) in lieu of validity calculations (*Van Hooren, Teratsias & Hodson-Tole, 2020*; *Kwah et al., 2013*). In *Van Hooren, Teratsias & Hodson-Tole (2020)*, the agreement between computational and manual ultrasound analyses was assessed with root mean square error (RMSE), which can be compared to the TE findings of the current study. For FA, RMSE was 1.02° to 22.96°, while FL was 8.25 to 14.32 mm. These findings are greater than the current study's TE for FA and FL. Without a consistent use of similar validity statics (*i.e.*, CE, TE, SEE) throughout the literature, the previous works are difficult to compare to the current study. However, these previous (*Van Hooren, Teratsias & Hodson-Tole, 2020*; *Kellis et al., 2009*) works do suggest ultrasound to be a valid technique to assess muscle architecture. While the automatic tool used in the present study is valid when compared to the manual method, future studies may wish to validate this technique against other imaging modalities (*i.e.*, MRI) and using different ultrasound operators (*Carr et al., 2021*).

As mentioned previously, many studies use various ultrasound frequencies. In medical imaging, ultrasound frequency is inversely proportional to its ability to penetrate tissue (*Edelman, 2012*). Said another way, when imaging of deep anatomical structures needed, investigators may benefit from using low-frequency ultrasound (*e.g.*, 8–10 MHz). Conversely, higher frequency may produce higher quality ultrasound images when examining superficial tissues (*Abu-Zidan, Hefny & Corr, 2011*). The findings of the current study are incongruous between the lower frequency and higher frequency analyses completed with the automated tool. For example, lower frequency analyses overall had larger error than the higher frequency analyses. Perhaps, when considered with the use of superficial images in the present study, the differences in accuracy between lower and higher frequency automated image analyses in the current study is the result of higher frequency causing less contrast between tissues and overall poorer image quality; therefore, leading to more error and poorer reliability compared to the lower frequency analyses. In support of this argument, the authors of the SMA program discuss that the algorithm uses contrast and echogenicity to distinguish between tissues (*Seynnes & Cronin, 2020*). Future studies may wish to employ a larger range of ultrasound frequencies across different depths of tissue to better understand the effect of hardware settings on image analysis error.

## Limitations

In the present study, it could not be ensured that the fascicle selected for the manual and automatic technique was identical nor that either technique assessed potentially curved fascicles without bias. Also, two separate data collection trials were needed to calculate the test-retest reliability statistics for the current study; therefore, it cannot be guaranteed that the same fascicle and its fascicle angle were captured and analyzed in both trials.

The present work utilized a single, unblinded investigator who analyzed all ultrasound images. Future investigators may wish to consider the effect of multiple raters on the reliability and validity statistic presented. While averaging fascicles lengths and fascicle angles may reduce the variability of the measure, the automated program used in the current study only selected a single fascicle length and angle to measure. Therefore, the authors manually measured a single fascicle and corresponding angle. Future investigators and developers of automated tools may wish to average multiple fascicles for their analyses in order to reduce variability. Lastly, the SMA macro procedure produced some muscle architecture measurements which may not be physiological (*e.g.*, FA > 60°). Future investigations of automated image processing techniques may benefit with the inclusion of a "filter" so that outlying fascicle angles would be removed. However, these values are left in the final dataset of the present investigation to maximize the current applicability of the SMA algorithm for the reader.

# CONCLUSION

In summary, the primary outcomes of this study indicated that manual ultrasound analyses were more reliable in comparison to the novel, automatic ultrasound analysis tool used in the present study in the calculation of FA and FL. It is recommended that the analyst is cautious when using the automated technique to analyze images taken at a higher frequency as the results indicated poor reliability for FA and FL. In addition to revealing good reliability, the manual ultrasound analyses presented less absolute error than the automatic analyses. When accuracy was assessed against manual analyses, the automatic tool presented moderate CE, TE, and SEE. These findings may be impactful as they highlight the good reliability and low absolute error associated with manually analyzed ultrasound images as well as the validation of a novel automatic tool for analyzing ultrasound images. Though promising, the reader is encouraged to employ caution in the use of automatic analysis tools for image processing until further work can be completed.

## Funding

The authors received no funding for this work.

## Competing Interests

The authors declare that they have no competing interests.

## Author Contributions

- Kealey J. Wohlgemuth conceived and designed the experiments, performed the experiments, analyzed the data, prepared figures and/or tables, authored or reviewed drafts of the article, and approved the final draft.
- Malia N. M. Blue conceived and designed the experiments, analyzed the data, prepared figures and/or tables, authored or reviewed drafts of the article, and approved the final draft.
- Jacob A. Mota conceived and designed the experiments, performed the experiments, analyzed the data, prepared figures and/or tables, authored or reviewed drafts of the article, and approved the final draft.

## Human Ethics

The following information was supplied relating to ethical approvals (*i.e.*, approving body and any reference numbers):

The University of Alabama granted ethical approval to carry out the study within its facilities (Institutional Review Board:#21-02-4385).

## Data Availability

The raw data are available in the Supplemental File.

## Supplemental Information

Supplemental information for this article can be found online at http://dx.doi.org/10.7717/peerj.13609#supplemental-information.

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
