# Peer review of "Reliability and accuracy of ultrasound image analyses completed manually versus an automated tool"

_PeerJ, doi:10.7717/peerj.13609_

## Round 0.1 · original submission · Major Revisions

The reviewers offer useful recommendations on your manuscript which if addressed will improve its readability, and its subsequent impact. Could you please address their comments. In addition please consider the following observations,

- The manuscript focusses on reliability and validity; the concept of validity from a statistical perspective is not well established in the literature, but reliability is. In the manuscript what is meant by validity? Is this not simply accuracy?

- The manuscript reports measures of muscle architecture, with the measure of the of the number of sarcomeres in parallel identified as pennation angle, but this is really a proxy would muscle thickness be a better measure?

·

Basic reporting

Overall the article is well written and able to be understood. I would add one thing to help clarify the abstract slightly. In the methods the low and high frequency images are explained (probe frequency) however, this is not explained in the abstract which can be confusing. Perhaps just noting it is probe frequency is changing would help.

Experimental design

Overall, the methods are fine. The one issue I had was with the repeated measures stats. It was difficult to understand what measure was repeated. The authors have a pre and a post measure in their data tables and they have two probe frequencies and two scans with a 10 minute break in between. Were the repeated measures between the first and seconds scans (with the 10 minute break in between) or were they between measurements where the individual analyzing the data analyzed the same image twice? If the pre post is between the two scans then I am very concerned about measuring the same fascicle each time. Previous research (Infantolino and Challis, 2014) has shown pennation variability within one image so this variability alone could explain the pre-post differences. If the scans change one would imagine this would become worse. Perhaps measuring multiple pennation angles per scan and averaging would help to reduce this issue.

Validity of the findings

Overall the findings are valid based on the stats. I would reiterate the issue I mentioned in the methods as a possible issue with the results in general. The other thing I noticed while looking at the raw data is that the automatically measured pennation angles were in many cases well outside the realm of reported pennation angles in the literature (>60 degrees). Is it possible to constrain the computer program to only look within a range of pennation angles (<60 degrees)? If so, would this greatly influence the accuracy of the results?

Additional comments

I like the idea behind this study and agree it is very important work. My two issues are the confusion with the repeated measures and the pennation angles that are well outside the realm of possibility.

Reviewer 2 ·

Basic reporting

General comments
Thank you for the opportunity to read and comment on the manuscript entitled “Reliability and validity of manual versus automatic ultrasound analyses”. The present study reports the reliability and validity of manual and automated methods to determine vastus lateralis architecture using B-mode ultrasonography. The manual analyses showed good reliability and the automatic analyses poor to moderate. However, absolute errors for both analyses were quite large, particularly in the automated analyses. The differences in outcome measures from the automated vs manual were very high and put into question the validity of the automated measurement program. These results are important for the field. Overall, the manuscript is well written, although there are sections requiring further improvement/clarification. I believe if they are met, it will form a much stronger manuscript. I also have some concerns regarding the methods used and more clarification on these are needed. Please refer to my specific comments below.

Specific comments
Tittle: It should be immediately clear that you’re testing the validity or previously published algorithms/automated analyses. I was under the impression that this was a newly developed coding/programming. This should be clear upfront.

Abstract: The authors should refer to fascicle angle instead of pennation angle. While pennation angle is generally used, it is conceptually wrong. Pennation angle is the angle between the fascicle and the muscle’s force line of action – tendon. Given that in pennate muscles, the aponeuroses are not necessarily in line with the tendon, we measure the angle of the fascicle instead of the angle of pennation. Please consider changing this throughout.
In the conclusion, I believe it’s better to provide a more general statement. What are the implications of these findings to the field? Why are they impactful? I also don’t see any inferences regarding the validity and the large errors in the automated analyses. Please consider including.

Introduction:
L91-92: This sentence is confusing. Please consider re-writing it

L94: this sentence is also unclear. Assessment of muscle architecture doesn’t directly imply number of sarcomeres in parallel and in series. It’s just that in pennate muscle we can fit more fascicles within the same space than a non-pennate muscle and that in lesser pennate muscles fascicles are often longer.

L94-97: Please revise the sentence.
L106-107: The authors suggest there is insufficient data regarding reliability and validity of automatic ultrasound analyses. I believe that published new automated analyses package/software are always validated at first glance. See Cronin et al 2020 (deep learning), Seynnes & Cronin 2020, Trackamte, UltraTrack (Farris & Litchwark 2016), etc. However, most of these manuscripts use single images with limited field of view, ie, they do not assess them using extended field of view.

L109: the authors suggest that only fascicle angle and length are part of the architectural measurements. However, muscle thickness is also very important and is strongly associated with maximal strength capacity and estimates of hypertrophy and its changes with training. I wonder why this has not been included in your analyses?

L110-11: there’s a problem with this statement. While it is generally accepted that automated analyses are compared against the manual ones to determine validity, this comparison will be highly dependent on the rater’s experience. If you have a sonographer that has no experience, the images will unlikely to be valid. But if the sonographer is highly experienced, it not only captures better, more valid, images but it also analyses them much better. However, one could also argue that the automated analyses from images obtained from both a non-experienced and experience sonographers and provides equally good/bad results, it will prove to be a valid coding/software. Haven said all of this, I what is the sonographer’s experience, how many images and analyses has the sonographer collected prior to this study, and why have you not included another rater to perform to capture the images and do the analyses? I also wonder if these analyses were blinded? All of that would have strengthen the methodological design of the paper.

In your hypothesis statement, can you provide an estimate of reliability beforehand? That, is, would you expect the reliability to be poor or good? And what sort of reliability are you referring to? This should be similarly applied to validity. These need further clarification….


Methods:
Why did the authors perform analyses in 23 participants only? Do you consider that enough for reliability and validity purposes? And why? Can you please justify your choice

Also, why was the study conducted on this specific population? If you’re determining the reliability and validity of a measurement, would you not want to test it in a population with very heterogenous characteristics? This would have allowed the results of the present study to be extrapolated to a broader range of population

The experimental design needs further information. I only understood that Trials 1 and 2 were used for reliability half-way through the methods. I think the sentence 124-125 is confusing: two independent data collection trials…. I think this could be improved to suggest you collected vastus lateralis images twice, 10 min apart, and that the images obtained from each time point were used to calculate fascicle angle and length and then estimate the within testing session reliability.

L126: Technically, ultrasound was used to capture images of the muscle of interest. The images allowed then for architecture to be calculated. Be clear.

L139: is this a secondary aim? Why was it done and why is it important? Please clarify that in the introduction. The reader won’t know why this was done.

L142-143: Two images per trial? So what did you do - took the average of fascicle angle and length for both? Sorry this is unclear

L143-144: What does “outer borders of the muscle” mean? The upper and lower apo? And what if you clearly see them but your probe is off angle and therefore fascicles aren't clearly delineated? I’d assume the main interest here is to see “fascicles” and therefore you wanted to guarantee fascicles were clearly visible.

L144: Please refer to the ultrasound images provided here.

L153-155: I'm concerned with this - vastus lateralis fascicles are not straight - they are curved and therefore you would be adding noise to your measurement if a straight line was considered. This will influence the validity of your measurement and affected the automated analyses. Please refer to Noorkoiv et al (2010, JAP).

L163: How is your manuscript different form the authors of this macro – ref 21? They showed validity of their automated programming in the ref 21

L171-172: If the automated program cannot identify fascicle angle and length from your "best" chosen image, it suggests that either (1) your image had poor quality or (2) the software doesn't do what it should be doing. Please clarify on this. It is also important to report how many times this occurred.

Statistical analyses:
L177: On what basis was this ICC chosen?

L178-179 & 181 & 185-189: Clarify how all the reliability measurements were calculated. The reader doesn't need to check Weir's paper to know how these were calculated. This also applies to the validity statistics.

L187: Why trial 1 only? Why didn't you repeat these to increase the number of comparisons to validate the automated analyses?

L193: ICC of 0.75 and SEM and MD of 14% and 7.4 degrees were reported. Does this suggest that, if an acute intervention was performed, you would need to change fascicle angle for at least 7.4 degrees for a change to be considered real? This is very impressive and practically impossible - changes in fascicle angle after fatiguing contractions or stretching are ~2.5 degrees. Do your data suggest they are possibly not real changes?
This should also be discussed further in the Discussion section.

L204: Somewhere it should be mentioned that the results show moderate reliability but very high errors!
L214-215: Do you consider these errors acceptable? They seem very high

L224: Can you speculate why the validity is worse for different frequencies?

Based on your figures, it seems to me that the only variable showing proportional bias is Figure A. Lower Frequency fascicle angle. All the others seem to increase the bias with increases in values.

Discussion:

L230: You need to make clear here that this is not your coding/program but that it has been published before. I suggest "...novel, previously published, automatic...."

L233-236: Please also refer to the SEM and other absolute reliability for this interpretation. ICCs only suggested that all subjects within the sample varied proportionally and therefore provides good between-subjects reliability; however, the absolute reliability is poor considering the expected changes in architecture with acute and chronic interventions.... this needs to be clarified.

L238: Tell the reader what TQ and SEE means.

L241-242: What do they add to? What's the implications of these findings? Please expand on this

L246-247: is this low? Which variables are these?

L249: The minimum provided here is significantly lower than the minimum SEM provided by the present study. Please discuss.

L251: Does this fit with the criterion adopted here? 0.63 does not seem an excellent score reliability.

L255-256: These errors are large (19-31%). This is within the range of muscle architectural changes with training and detraining.
L2625-263: Why did you only analysed one fascicle? If this information was known prior to the study being performed, then you would expect to do something similar so you can discuss the findings to that specific study.

L264-266: Why didn't the authors expand further and analysed multiple fascicles? If this is indeed quick, it shouldn't have taken longer to analyse another fascicle of the same image. Please justify.

L269-271: This seems an unacceptable error and invalidates the automated program. On top of these differences between manual and automated analyses, you are also expecting a big between trial variation, further reassuring the inability of the automated program to accurately and reliability determine fascicle length and angle.

L274: What is a true validity calculation? Please be clear

L281-283: Why have you not performed those calculations to make it, at the least, comparable with the literature?

L289-293: Why were they not identifiable? If the images are performed under the same conditions, by the same experience investigator, would you not be able to place an image side by side and tell whether the region of the fascicles chosen are matched and therefore select the same fascicle? I also think analyses of a single fascicle is complicated. You may need to include that in your limitations. How can you guarantee the fascicle chosen was representative of the fascicles of the muscle?

L303-304: If the errors were large, would you consider this valid? The concept of validity needs to be clarified.

L306: Theres no mention of how these are applied to training, detraining, or acute interventions. A recent paper has shown that the changes in muscle architecture after disuse are method dependent (Sarto et al 2021 - 10.1249/MSS.0000000000002614). Considering the large errors showed here, this should be discussed!

Conclusion
L298-299: Can you please provide an explanation for this? Can you speculate as to why this is impacting on your measurements?

References:
Please check your reference throughout so that tittles are standardized. See ref 3 vs 2 - tittle has capital letters in each word whereas the other doesn’t.

Figures/Tables
Figure 1: If A and C are the same participant, there are clear differences in where the fascicle starts and ends and how the aponeuroses are determined – this is important because it significantly affects fascicle length and angle measurements.
It is unclear what the boxes in A and B are.

Figure 2: The x-axis of Figures A-D should be the average of the variable name, e.g., “average fascicle angle (degrees)”. Additionally, on the top of the figure, can you please write the variable name and in parentheses the US frequency?

Table 2: since it's unclear to me how you calculated any of these validity outcomes, it's hard to tell why this is in cm. The total error a can also be calculated as % value or arbitrary units, depending on how it is determined. Please clarify.

Experimental design

Full report is above.

Validity of the findings

N/A

---

## Round 0.2 · Minor Revisions

The comments to address are in the attached PDF.

·

Basic reporting

The reviewer thanks the authors for the responses to my comments. The clarifications have helped with my understanding of the article.

Experimental design

I understand why you chose to not use an average fascicle value. I am not thrilled with the idea but I can understand why you went in this direction.

Validity of the findings

I understand the constraints of the SMA macro produced results that were non-physiologic. The additional wording you added helps to explain this. While I wish that were not the case, I understand you are limited by the program you were testing.

Reviewer 2 ·

Basic reporting

The authors did a great job in addressing the questions and significantly improved the article. I have only provided minor comments/suggestions to further improve the manuscript. I have addressed them in blue throuhgout the word document attached.

Experimental design

N/A

Validity of the findings

N/A

Annotated reviews are not available for download in order to protect the identity of reviewers who chose to remain anonymous.

---

## Round 0.3 · accepted · Accept

Thank you for addressing the previous round of comments.